# PEGylated Polyurea Bearing Hindered Urea Bond for Drug Delivery

**DOI:** 10.3390/molecules24081538

**Published:** 2019-04-18

**Authors:** Meishan Chen, Xiangru Feng, Weiguo Xu, Yanqiao Wang, Yanan Yang, Zhongyu Jiang, Jianxun Ding

**Affiliations:** 1Chemical Engineering Institute, Changchun University of Technology, 2055 Yan’an Street, Changchun 130012, China; mschen@ciac.ac.cn; 2Key Laboratory of Polymer Ecomaterials, Changchun Institute of Applied Chemistry, Chinese Academy of Sciences, 5625 Renmin Street, Changchun 130022, China; xrfeng@ciac.ac.cn (X.F.); wgxu@caic.ac.cn (W.X.); candyqiao77@jlu.edu.cn (Y.W.); jxding@ciac.ac.cn (J.D.); 3Jilin Biomedical Polymers Engineering Laboratory, 5625 Renmin Street, Changchun 130022, China

**Keywords:** amphiphilic copolymer, hydrolyzable polyurea, micelle, controlled drug delivery, cancer chemotherapy

## Abstract

In recent years, polyureas with dynamic hindered urea bonds (HUBs), a class of promising biomedical polymers, have attracted wide attention as a result of their controlled hydrolytic properties. The effect of the chemical structures on the properties of polyureas and their assemblies has rarely been reported. In this study, four kinds of polyureas with different chemical groups have been synthesized, and the polyureas from cyclohexyl diisocyanate and *tert*-butyl diamine showed the fastest hydrolytic rate. The amphiphilic polyurea composed of hydrophobic cyclohexyl-*tert*-butyl polyurea and hydrophilic poly(ethylene glycol) (PEG) was synthesized for the controlled delivery of the antitumor drug paclitaxel (PTX). The PTX-loaded PEGylated polyurea micelle more effectively entered into the murine breast cancer 4T1 cells and inhibited the corresponding tumor growth in vitro and in vivo. Therefore, the PEGylated polyurea with adjustable degradation might be a promising polymer matrix for drug delivery.

## 1. Introduction

Polyureas bearing hindered urea bonds (HUBs) are able to be synthesized by the reaction between the monomers with diisocyanate or diamine groups [1,2]. As the orbital coplanarity of the amide bonds can be disturbed [3], HUBs are hydrolyzable, and the hydrolytic products are biologically safe [4]. Furthermore, the hydrolyzability can be regulated by changing the units of polyureas [5]. Due to their excellent hydrolyzability and biocompatibility, polyureas have been developed for biomedical applications, including drug delivery [6,7,8].

As drug carriers, the drug release profiles of polyureas can be adjusted by changing the hydrolytic rates [4], and the accumulation of therapeutic drugs at the desired tissues can also be enhanced by passive and active targeting. For instance, Shoaib et al. developed several polyurethane-urea elastomers with various diisocyanate groups to deliver the antitumor drug doxorubicin (DOX). The drug release could be adjusted by changing the pH of media and hard segment chemical structures of the polyureas [9]. Morral-Ruiz et al. developed the biotinylated polyurethane-urea nanoparticles for target delivery of plasmids and drugs [10]. The nanoparticles were loaded with a reporter gene containing plasmids and antitumor drugs for simultaneous theranostics of cancer cells. The nanoplatforms based on biotinylated polyurethane-ureas were potentially used for the therapy of other types of cancer. John et al. synthesized a series of polyureas with disulfide linkages in the backbone, and the antitumor drug DOX was encapsulated by the nanocarriers [7]. The drug-loaded nanocarriers demonstrated the glutathione-responsive drug release behavior, indicating significant potential for controlled drug delivery. Although polyureas have been widely explored as a nanocarrier due to their biological safety and modifiability, the effect of the chemical structures of HUBs on the properties of hydrolyzable polyureas as a drug nanocarrier has rarely been reported.

In our study, a series of hydrolyzable polyureas with different chemical structures were synthesized through different diisocyanate monomers (i.e., cyclohexyl and benzyl diisocyanates) and diamine monomers (i.e., isopropyl, *tert*-butyl, and hydroxyethyl diamines). The polyureas were denoted as poly(**1**/**3**), poly(**1**/**4**), poly(**2**/**3**), and poly(**2**/**5**), as depicted in Scheme 1. The four polyureas showed distinct hydrolytic rates. Among the obtained polyureas, poly(**1**/**3**), with the fastest hydrolytic property, had the most ideal potential as a nanocarrier for drug release. As the amphiphilic PEGylated polyurea, methoxy poly(ethylene glycol)−poly(**1**/**3**) (mPEG−poly(**1**/**3**)), composed of hydrophilic amino-terminated mPEG (mPEG_113_-NH_2_) and hydrophobic poly(**1**/**3**), was synthesized according to the reaction between the amino group of mPEG and the isocyanate group of polyurea. In addition, paclitaxel (PTX), a traditional hydrophobic antitumor agent, was loaded into mPEG−poly(**1**/**3**) to obtain the drug-loaded polyurea micelle (PUM/PTX), as shown in Scheme 2. PUM/PTX could be efficiently internalized by mouse breast cancer 4T1 cells and showed noticeable cytotoxicity in vitro. Moreover, PUM/PTX was also proven to be potent in inhibiting the growth of murine breast tumor compared to PTX treatment alone in vivo. Both histopathology and immunofluorescence were further conducted to validate the enhanced antitumor effect and biological safety of PUM/PTX. In conclusion, the PEGylated polyurea with adjustable degradation might be a meaningful polymer for drug delivery.

## 2. Results and Discussion

### 2.1. Syntheses and Characterizations

Four kinds of hydrolyzable polyureas with different chemical groups were successfully synthesized. Briefly, equal amounts (1.0 mmol) of each monomer (**1** and **3**, **1** and **4**, **2** and **3**, or **2** and **5**) were mixed in deuterated chloroform (CDCl_3_, 5.0 g). The solutions were vigorously stirred at room temperature overnight, two-fold diluted without purification, and directly characterized by proton nuclear magnetic resonance (^1^H NMR). All peaks in the ^1^H NMR spectra of polymers were accurately assigned (Figure 1a,b). The characteristic peaks at 5.87, 6.29, and 6.46 ppm were attributed to the hydrogen atom in HUBs. The typical resonances at 1.12–1.30 and 3.62–4.70 ppm represented the substituents on a nitrogen atom. Fourier-transform infrared (FT-IR) spectroscopy further confirmed the chemical structures of polyureas as the ^1^H NMR results. The typical signals of ureas were at 3300–3320 cm^−1^. The typical wavenumbers of PEG were at 1360, 1297, and 1249 cm^−1^. These results demonstrate that the hydrolyzable polyureas were successfully synthesized (Figure 1c,d).

The number-average molecular weights (*M*_n_s) of polyureas were obtained from gel permeation chromatography (GPC) at the scheduled time points (Figure 1e). It was observed that the hydrolytic rates of poly(**1**/**3**) and poly(**2**/**3**) in a mixed solution were faster than poly(**1**/**4**) and poly(**2**/**5**). After 48 h, *M*_n_s reduction percentages of poly(**1**/**3**), poly(**2**/**3**), poly(**1**/**4**), and poly(**2**/**5**) were 78.4%, 74.6%, 6.4%, and 4.8%, respectively. The hydrolytic rate of poly(**1**/**3**) increased to 82.5% after incubation for 72 h. The results could be explained as the bulky substituents incorporating into one of the nitrogen atoms [11]. Urea bonds could be easily destabilized by disarranging the orbital coplanarity of the amide bonds and diminished the conjugation effect. Urea bonds with a bulky substituent could dissociate into isocyanate and amines reversibly. Isocyanates could hydrolyze into amines and carbon dioxide (CO_2_) in an aqueous solution. It was an irreversible reaction that shifted the balance to aid the HUBs dissociation reaction and finally led to the complete hydrolysis of HUBs. Since poly(**1**/**3**) had a suitable hydrolytic time, it was selected for the following experiments.

In order to increase the hydrophilia of polyureas, mPEG was attached to both ends of poly(**1**/**3**). The ^1^H NMR spectrum of mPEG−poly(**1**/**3**) dissolved in CDCl_3_ demonstrated the successful synthesis of PEGylated polyurea (Figure 1a). In mPEG−poly(**1**/**3**), the mPEG segment was the hydrophilic segment, and the poly(**1**/**3**) moiety was the hydrophobic moiety. The amphiphilic mPEG−poly(**1**/**3**) self-assembled into micelle in phosphate-buffered saline (PBS). Observed by transmission electron microscopy (TEM; Figure 2a), the PUM showed a spherical structure with a mean radius of around 38 nm. The hydrodynamic radius (*R*_h_) of PUM determined by dynamic laser light scattering (DLS) was 39.6 ± 8.1 nm. The radius of PUM examined by TEM showed the similar result to that by DLS. In order to explore the hydrolytic characteristic of mPEG−poly(**1**/**3**), the *M*_n_s were detected at the scheduled time points. PUM exhibited the desired hydrolytic rates in PBS, which was similar to poly(**1**/**3**). After incubation for 48 h, the *M*_n_ of PEGylated PUM was reduced to 71.5% (Figure 2b).

The drug encapsulation capability is another important requirement for a suitable and robust drug delivery system [12,13,14]. In order to explore the drug loading properties of the polymer, PTX was loaded into mPEG−poly(**1**/**3**) micelle to obtain PUM/PTX. The drug loading content (DLC) and drug loading efficiency (DLE) were computed using the following Equations (1) and (2) [15,16]:
(1)$$\mathrm{DLC}\;=\;\frac{\mathrm{Weight}\;\mathrm{of}\;\mathrm{drug}\;\mathrm{in}\;\mathrm{PUM}/\mathrm{PTX}}{\mathrm{Weight}\;\mathrm{of}\;\mathrm{drug}-\mathrm{loaded}\;\mathrm{micelle}}\;\times \;100\%$$
(2)$$\mathrm{DLE}\;=\;\frac{\mathrm{Weight}\;\mathrm{of}\;\mathrm{drug}\;\mathrm{in}\;\mathrm{PUM}/\mathrm{PTX}}{\;\mathrm{Weight}\;\mathrm{of}\;\mathrm{feeding}\;\mathrm{drug}\;}\;\times \;100\%$$

PTX was successfully loaded into mPEG−poly(**1**/**3**) micelle with DLC of 8.7% and DLE of 87.5%. The morphological characteristic of PUM/PTX was observed by TEM (Figure 2c), and the average nanoparticle size was around 43 nm. The *R*_h_ was 44.7 ± 11.6 nm by the DLS test, showing a similar result with TEM. The results suggested that PUM/PTX was a monodisperse micelle and had a uniform particle size distribution.

### 2.2. PTX Release, In Vitro Cell Uptake, and Cell Proliferation Inhibition

The release characteristics of PTX from PUM/PTX were detected in PBS. As depicted in Figure 2d, the release profile showed no apparent burst release of PTX from PUM/PTX in PBS at pH 7.4 within 24 h. The amount of PTX released from PUM/PTX was lower than 45% during the first 12 h. At 48 h, over 65% of PTX was released. Since the tumoral microenvironment is more acidic, the PTX release behavior was also tested in PBS at pH 6.8 (Figure 2e). The amount of PTX released from PUM/PTX was 46% in the first 12 h. At 60 h, about 69% of PTX was released. These two release rates indicated that the hydrolytic rate of PUM was faster in the acidic condition. This controlled release pattern indicated that polyurea could be used as a suitable drug delivery system.

Drug release was a complicated process [13]. To just explain the nature of drug release behaviors, a classic empirical equation was established by Peppas et al. [17]. The equations were written as:
(3)$$\frac{M_{t}}{M_{\infty }}{\;=\;k\;t}^{n}$$
(4)$$\mathrm{lg}\;\left(\frac{M_{t}}{M_{\infty }}\right)\;=\;\mathrm{lg}\;k\;+\;n\;\mathrm{lg}\;t\;$$

In Equations (3) and (4), *M_t_* and *M*_∞_ were the cumulative drug release at time *t* and infinite time, respectively; *k* was the proportionality constant, and *n* was the release exponent that was related to the release mechanism of payloads. In the study of drug release, an increase in *n* indicates that the release was more influenced in a swelling-controlled way. *n* was calculated using the Equations (3) and (4), and the values for a pH of 6.8 and 7.4 were 0.32 and 0.21, respectively. The value of *n* was larger at pH 6.8 than pH 7.4, which was attributed to the faster hydrolysis of polyurea in an acidic environment.

Coumarin-6 (C_6_) as a hydrophobic model fluorescence molecule was used for cell uptake study [18]. C_6_ was loaded into mPEG−poly(**1**/**3**) micelle to form PUM/C_6_. The internalization of PUM/C_6_ by 4T1 cells was monitored through flow cytometry (FCM) and confocal laser scanning microscopy (CLSM). As shown in Figure 3a, after 1 h incubation, the control group had no fluorescence signal of C_6_, while in the PUM/C_6_ group the fluorescence signals were significantly increased, and fluorescence intensity was further increased at 6 h. Similarly, CLSM images in Figure 3b,c showed that the highest fluorescence intensity was detected when PUM/C_6_ was incubated with 4T1 cells for 6 h. The results demonstrated that the PUM micelle could efficiently deliver PTX into 4T1 cells.

PUM with HUBs was synthesized by the reversible reaction between cyclohexyl diisocyanate and *tert*-butyl diamine. With the hydrolysis of PUM, the degradation products were cyclohexane-1,3-diyldimethanamine and *tert*-butyl diamine. The cytotoxicity of the hydrolytic products of HUBs to 4T1 and L929 cells was tested by methyl thiazolyl tetrazolium (MTT) assays (Figure 4a, b). After incubation with PUM at the concentration of 100.0 μg mL^−^^1^ for 72 h, the viability of L929 cells was kept around 93%, indicating negligible toxicity of PUM to normal cells. After incubation with PUM for 72 h at a concentration of 50.0 μg mL^−1^, the cell viability of 4T1 cells was 86.97%, indicating that the polymer had little effect on the growth of tumor cells. MTT assays compared the toxicity of free PTX and PUM/PTX toward 4T1 cells. Both free PTX and PUM/PTX inhibited the growth of 4T1 cells. The cell viability was reduced to 51.9% after incubation with free PTX for 48 h at a concentration of 10.0 μg mL^−1^, while the cells treated with an equivalent dose of PUM/PTX showed viability of 46.6% (Figure 4c). As shown in Figure 4d, the cell proliferation was further suppressed at 72 h with cell viability reduced to around 45.9%. The data above confirmed that PUM/PTX could be efficiently endocytosed by 4T1 cells and release PTX to perform the antitumor effect. In vitro experiments proved that the polymer could make a proper drug delivery vehicle.

### 2.3. In Vivo Antitumor Efficacy

The antitumor activity was investigated in a BALB/c model of mice bearing allograft orthotopic murine 4T1 breast tumors. The antitumor efficacy of PUM/PTX was further detected in vivo. Free PTX was dissolved in the mixture of castor oil and ethanol [19], and diluted with PBS. The mice were treated with free PTX or PUM/PTX at a dosage of 5.0 mg kg^−1^ PTX, and the mice treated with PBS were set as a control group. The treatment started at the time when the tumor volumes reached approximately 200 mm^3^. As shown in Figure 5a, free PTX showed modest antitumor efficacy compared with the control group within 18 days, whereas PUM/PTX showed the best inhibition rate of 32.7%. This result suggests that PUM/PTX could well inhibit tumor growth. The photograph of the tumors had a similar result (Figure 5b).

Histopathology and immunofluorescence analyses were further conducted on the isolated tumors to evaluate the antitumor efficacies of PBS as control, free PTX, and PUM/PTX. In this work, lung metastasis, the trauma to healthy tissue, and the antitumor efficacies of all groups were evaluated by hematoxylin and eosin (H&E) staining. As shown in Figure 5c, without the appearance of lung metastasis, intratumoral administration of PTX and PUM/PTX at a dose of 5.0 mg kg^−1^ did not cause a noticeable change in the histopathological morphologies of the lungs and spleens. In the PTX and PUM/PTX groups, tumor cells showed a decrease in volume and denser cytoplasm compared with those of the control group. A large area of apoptosis appeared in the tumor tissue. The H&E results were further verified by immunofluorescence staining of Ki-67 and caspase-3. Immunofluorescence staining of Ki-67 was used to evaluate the effect of PTX formulations on the growth of tumor [20]. As shown in Figure 5d, both PUM/PTX and free PTX-treated group showed less Ki-67-positive cells. Especially, a slightly stronger antitumor effect was observed in the PUM/PTX group. The caspase-3 analysis is a standard method to analyze the apoptosis of tumor cells [21]. The amount of caspase-3 suggests the apoptosis level of cells in different tumor tissues. Compared with the control group, more strong signals of caspase-3 were manifested in tumor tissues of the free PTX and PUM/PTX group, indicating a larger apoptosis area (Figure 5e). These results were consistent with histopathological H&E analyses.

The body weight is an essential physiological factor to assess the toxicity of drugs [22]. As exhibited in Figure 5f, mice treated with free PTX showed an apparent decrease in body weight compared with the control group, while the body weight of mice in the PUM/PTX group remained stable. This phenomenon indicates that mPEG−poly(**1**/**3**) could efficiently improve safety. All experiments validated that mPEG−poly(**1**/**3**) could act as a suitable and biological safety drug delivery vehicle.

## 3. Materials and Methods

### 3.1. Materials

1,3-Bis(isocyanatomethyl) cyclohexane (**1**) and 3-bis(isocyanatomethyl) benzene (**2**) were purchased from Tokyo Chemical Industry Co., Ltd. (Shanghai, China), *N,N’*-di-*tert*-butyl-ethylenediamine (**3**) was purchased from Biological Science and Technology Co., Ltd. (Shanghai, China), and *N,N’*-di-*iso*-propylethylenediamine (**4**) was purchased from Aladdin (Shanghai, China), and all of them were used as obtained. *N,N’*-Bis(2-hydroxyethyl)ethylenediamine (**5**) was purchased from Energy Chemical (Shanghai, China). *N,N*-dimethylformamide (DMF) and ethyl ether were bought from Tiantai Fine Chemical Co., Ltd. (Tianjin, China). DMF was stored over calcium hydride (CaH_2_) and purified by vacuum distillation. C_6_ and mPEG_113_ were purchased from Sigma-Aldrich (Shanghai, China). mPEG_113_−NH_2_ was prepared as the protocol reported in our previous work [23]. PTX was purchased from Huafeng United Technology Co., Ltd. (Beijing, China). Dulbecco’s modified Eagle’s medium (DMEM) and newborn bovine serum (NBS) were bought from Gibco (Grand Island, NY, USA) and Every Green (Hangzhou China), respectively. Methyl thiazolyl tetrazolium (MTT) and 4’,6-diamidino-2-phenylindole dihydrochloride (DAPI) were purchased from Sigma-Aldrich (Shanghai, China). The primary antibody was purchased from Abcam Company (Cambridge, UK). The secondary antibody was purchased from ABclonal (Wuhan, China). The purified deionized water was prepared by the Milli-Q plus system (Millipore Co., Billerica, MA, USA).

### 3.2. Syntheses of Four Different Polyureas

The equimolar of **1** (1.94 g, 10.0 mmol) and **3** (1.72 g, 10.0 mmol), **1** (1.94 g, 10.0 mmol) and **4** (1.44 g, 10.0 mmol), **2** (1.88 g, 10.0 mmol) and **5** (1.48 g, 10.0 mmol), **2** (1.88 g, 10.0 mmol) and **3** (1.72 g, 10.0 mmol) were separately dissolved in anhydrous DMF (10.0 g). The solutions were vigorously stirred at room temperature overnight. The polymer solutions had 5% water added to them, and they were then shaken at 80 rpm in a 37 °C incubator. Then they were used for the study of hydrolysis directly [5].

### 3.3. Synthesis of mPEG−poly(1/3)

The **1** (1.3 g, 6.7 mmol) and **3** (1.1 g, 6.5 mmol) were dissolved in anhydrous DMF, separately, and vigorously stirred at room temperature overnight. mPEG_113_-NH_2_ (2.5 g, 0.5 mmol) was dissolved in toluene, and residual water in the solution was removed by azeotropic distillation. The dehydrated mPEG_113_-NH_2_ was dissolved in anhydrous DMF, and then added into the reaction flask. The mixture was stirred at room temperature for three days and was precipitated by anhydrous ethyl ether.

### 3.4. Preparation of PUM

mPEG−poly(**1**/**3**) was dissolved in dimethyl sulfoxide (DMSO) and slowly added into 0 °C PBS. The mixture solution was ultrafiltrated by an ultrafiltration tube (molecular weight cut-off (MWCO) = 10,000 Da; Millipore Co., Billerica, MA, USA). Finally, the concentration of mPEG−poly(**1**/**3**) solution was kept at 1.0 mg mL^−1^.

To investigate the hydrolysis of mPEG−poly(**1**/**3**), PUM was shaken at 80 rpm in a 37 °C incubator, 3.0 mL of the liquid was pipetted at different time points, and the liquid was lyophilized. Finally, the lyophilized solids were dissolved in DMF, and *M*_n_s were monitored by GPC.

### 3.5. Preparation of PUM/PTX

First, mPEG−poly(**1**/**3**) and PTX were dissolved in DMSO with a mass ratio of 9:1. Then, the two solutions were mixed evenly. The mixture was added into 0 °C PBS slowly while stirring quickly. Finally, the DMSO of the mixture solution was removed by ultrafiltration.

### 3.6. Characterizations

^1^H NMR spectra were recorded on a Bruker AV 400 NMR spectrometer in CDCl_3_. FT-IR spectra were recorded on a Bio-Rad Win-IR instrument (Cambridge, MA, USA). GPC analyses of polymers were conducted on a Waters 2414 system (Waters Co., Milford, MA, USA) equipped with Ultrahydrogel™ linear columns and a Waters 2414 refractive index detector (injection volume: 30.0 μL, column temperature: 50 °C, eluant: DMF through 0.45 and 0.22 μm filters, flow rate: 1.0 mL min^−1^). TEM measurements were performed on a JEOL JEM-1011 transmission electron microscope with an accelerating voltage of 100 kV. DLS measurements were performed with a vertically polarized He−Ne laser (DAWN EOS, Wyatt Technology Co., Santa Barbara, CA, USA). The high-performance liquid chromatography (HPLC) analyses of PTX were performed with a Symmetry^®^ C18 column connected to a Waters 2487 (Waters Co., Milford, MA, USA) at a flow rate of 1.0 mL min^−1^. The DLC and DLE of PUM/PTX were determined according to the following protocol. A volume of 200.0 μL of the PUM/PTX sample was mixed into 800.0 μL of acetonitrile, and the detection wavelength was 227 nm.

### 3.7. In Vitro PTX Release

The PUM/PTX solution was placed into dialysis bags (MWCO = 3500 Da). The dialysis bags were transferred into PBS at pH 6.8 and 7.4, 37 °C with 80 rpm. HPLC tests detected the amount of released PTX.

### 3.8. Cell Cultures

4T1 cells were cultured in complete DMEM, supplemented with 10% (*V*/*V*) NBS, penicillin (50.0 IU mL^−1^), and streptomycin (50.0 IU mL^−1^) at 37 °C in a 5% (*V*/*V*) CO_2_ atmosphere.

### 3.9. Cell Uptakes

PUM/C_6_ was used for the cell uptake study. Both CLSM and FCM investigated the cell uptake of PUM/C_6_ toward 4T1 cells.

#### 3.9.1. FCM

The cell uptake study was conducted with C_6_, which was used as a hydrophobic model fluorescence probe. The cells were seeded in 6-well plates at a density of 2.0 × 10^5^ cells to each well and cultured for 12 h. Then, 100.0 μL of PUM/C_6_ was added into the wells, and the cells were further incubated at 37 °C for 1 or 6 h. Next, the harvested cells were suspended in PBS and centrifuged at 1,000 rpm for 5 min. All the cells were washed with PBS. Finally, the cells were resuspended with 500.0 μL of PBS. Data were analyzed by FCM on a Guava^®^ easyCyte^TM^ flow cytometer (Merck Millipore, Darmstadt, Germany).

#### 3.9.2. CLSM

The cells were seeded on the coverslips in 6-well plates with a density of 2.0 × 10^5^ cells/well. A volume of 2.0 mL of DMEM was added into the cells per well and cultured for 12 h. Then 100.0 μL of PUM/C_6_ was added to each well. After incubation for 1 or 6 h, the cells were fixed with 4% (*W*/*V*) formaldehyde for 15 min. Then, the fixed cells were incubated with DAPI for 3 min and washed with PBS. The images of cell localization were observed under LSM 780 CLSM (Carl Zeiss, Jena, Germany) with 10× eyepiece and 40× objective.

### 3.10. Cytotoxicity Assays

The MTT assays evaluated the cytotoxicity of PUM/PTX. 4T1 cells with a density of 5.0 × 10^3^ cells/well were seeded in 96-well plates in 180.0 μL of DMEM and incubated for 24 h. A volume of 20.0 μL of free PTX or PUM/PTX was added in each well with a maximum PTX concentration of 200.0 μg mL^−1^. The cells were subjected to MTT assay after being incubated for another 48 or 72 h. The absorbance of the above solution was measured on a Bio-Rad 680 microplate reader (Hercules, CA, USA) at 490 nm. The cells viability was calculated based on the following Equation (5):(5)$$\mathrm{Cells}\;\mathrm{viability}\;(\%)\;=\;\frac{A_{\mathrm{sample}}}{A_{\mathrm{control}}}\;\times \;100\%$$

In Equation (5), *A*_sample_ was denoted as the absorbance of the sample, and *A*_control_ was denoted as the absorbance of the control.

The cytotoxicity of PUM to 4T1 and L929 cells was also tested at 72 h. The specific steps were the same as above.

### 3.11. Animal Procedures

Female BALB/c mice at five weeks of age were obtained from Vital River Laboratory Animal Technology Co., Ltd. (Beijing, China). The body weight of the mice was kept between 18 to 20 g before the start of the experiment. The experiments on the animals were carried out according to the guidelines outlined in the Guide for the Care and Use of Laboratory Animals, provided by Jilin University (Changchun, China) and the procedures were approved by the Animal Care and Use Committee of Jilin University (Protocol No. 2017-154).

### 3.12. In Vivo Antitumor Efficacy

BALB/c mice were inoculated with 4T1 cells to develop the breast tumor xenograft model. The breasts were injected with 1.0 × 10^6^ cells in 0.1 mL of PBS. The mice were treated with PBS, PUM/PTX, or free PTX on day 0, 4, 8, 12, and 16 through intratumoral injections. The tumor volumes (*V*, mm^3^) were estimated using the following Equation (6) [24],
(6)$$V\;({\mathrm{mm}}^{3})\;=\;\frac{{a\;\times \;b}^{2}}{2}$$

In Equation (6), *a* and *b* (mm) were the largest and the smallest axes of the tumor, measured by a caliper.

### 3.13. Histopathological and Immunofluorescence Analyses

The mice were sacrificed two days after the last injection. According to the protocol reported in previous studies, the lung is the most common organ for breast cancer metastasis, while the spleen is the primary immune organ that should be observed at the end of the experiments [25]. The tumors and major organs (i.e., the lungs and spleens) were collected, fixed in 4% (*W*/*V*) PBS-buffered paraformaldehyde overnight, and then embedded in paraffin. The paraffin-embedded tissues were cut into ~5 μm slices for H&E staining and ~3 μm sheets for immunofluorescence analyses (i.e., Ki-67 and caspase-3). The histological and immunofluorescence alterations were detected by a microscope (Nikon Eclipse Ti, Optical Apparatus Co., Ardmore, PA, USA).

## 4. Conclusions

In this study, we synthesized four kinds of hydrolyzable polyureas with different hydrolytic rates by changing the chemical groups on the polyureas. Among them, poly(**1/3**) from cyclohexyl diisocyanate and *tert*-butyl diamine showed the fastest hydrolytic rate. After modification by hydrophilic mPEG, the amphiphilic mPEG−poly(**1**/**3**) was synthesized for delivery of PTX. The PTX was successfully encapsulated by mPEG−poly(**1**/**3**) micelle with a DLC and DLE of 8.75% and 87.5%, respectively. PUM/PTX could be efficiently internalized by murine breast cancer 4T1 cells and released PTX along with the hydrolysis of polyurea. The results showed that PUM/PTX drastically suppressed the proliferation of tumor cells in vitro and significantly inhibited tumor growth in an orthotopic 4T1 breast tumor model in vivo. Therefore, the hydrolyzable PEGylated polyureas with adjustable degradation might become a promising platform for controlled drug delivery.

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
