# Peer review of "PEGylated Polyurea Bearing Hindered Urea Bond for Drug Delivery"

_molecules, 2019, doi:10.3390/molecules24081538_

Round 1

Reviewer 1 Report

The manuscript entitled “PEGylated Polyurea Bearing Hindered Urea Bond for Drug Delivery” is well written and organized. However some considerations need to be performed before publication.

1. In order to better characterize the hydrolysable polyureas I recommend the authors to carry out FTIR spectroscopy, to confirm the H-NMR results.

2. In the release studies, I recommend the authors the application of mathematical models (Higuchi, Korsmeyer Peppas, First order,…) in order to evaluate the mechanism of drug release from the micellar systems.

3. Which pH the authors used in the PBS to perform release studies? If authors used pH=7.4, I remember that the tumoral microenvironment is more acidic.

4. In the cytotoxicity studies I recommend the authors to evaluate the cytotoxicity of the hydrolysable polyureas in healthy cells.

Author Response

The manuscript entitled “PEGylated Polyurea Bearing Hindered Urea Bond for Drug Delivery” is well written and organized. However some considerations need to be performed before publication.

Response: Thanks so much for your positive comments on our manuscript. All the issues arising from your comments have been well addressed and responded as follows.

1. In order to better characterize the hydrolysable polyureas I recommend the authors to carry out FTIR spectroscopy, to confirm the H-NMR results.

Response: Thanks very much for your constructive advice. FT-IR spectroscopy was supplemented in the revised manuscript, in order to confirm the 1H NMR results. We added the following description:

“Fourier-transform infrared (FT-IR) spectroscopy further confirmed the chemical structures of polyureas as the 1H NMR results. The typical signals of ureas were at 3300 − 3320 cm−1. The typical wavenumbers of PEG were at 1360, 1297, and 1249 cm−1. These results demonstrated that the hydrolyzable polyureas were successfully synthesized (Figure 1c, d)."

(c)

(d)

2. In the release studies, I recommend the authors the application of mathematical models (Higuchi, Korsmeyer Peppas, First order,…) in order to evaluate the mechanism of drug release from the micellar systems.

Response: Thanks a lot for your kind reminding. We added relevant content in the manuscript as follows:

“Drug release was a complex process [15]. To simply explain the nature of drug release behaviors, a classic empirical equation was established by Peppas et al. [16]. The eqs. were written as:

(3)

(4)

In eq (3) and (4), Mt and M were the cumulative drug release at time t and infinite time, respectively; k was the proportionality constant, and n was the release exponent that it was related to the release mechanism of payloads. In the study of drug release, the increase in n indicated that the release was more influenced by a swelling controlled way. n was calculated using the eqs. (3) and (4), and the values of pH 6.8 and 7.4 were 0.32 and 0.21, respectively. The values of n were larger at pH 6.8 than pH 7.4, which was  attributed to the faster hydrolysis of polyurea in an acidic environment.’’

3. Which pH the authors used in the PBS to perform release studies? If authors used pH=7.4, I remember that the tumoral microenvironment is more acidic

Response: Thank you for your reminding. After careful thought, we determined to add the release studies in the PBS (pH 6.8), and supplemented as follow:

“Since the tumoral microenvironment is more acidic, the PTX release behavior was also tested in PBS at pH 6.8 (Figure 2e). The amount of PTX released from PUM/PTX was 46% at the first 12 h. At 60 h, about 69% of PTX was released. These two release rates indicated that the hydrolytic rate of PUM was faster in the acidic condition.”

4. In the cytotoxicity studies I recommend the authors to evaluate the cytotoxicity of the hydrolysable polyureas in healthy cells.

Response: Thanks very much for your constructive comment. We have added the experiment that tested the cytotoxicity of the hydrolysable polyureas in normal cells. We added relevant content in the manuscript as follows:

“After incubation with PUM at the concentration of 100.0 μg mL–1 for 72 h, the viability of L929 cells was kept around 93%, 

Reviewer 2 Report

The study looks at the synthesis a of 4  hydrolyzable polyurea  with different chemical structures to enable drug delivery via diffusion . The molecules were synthesized by a reaction between the diisocyanate and diamine monomers and extensively characterized .The ability of one of the polymers to deliver paclitaxel (PTX) in a murine model, both in vivo and in vitro is studied. HUB-based polymers are a promising route for degradable biomaterials and the authors have well- characterized the synthesized monomers: however, there are few issues that need to be addressed before the paper can be published

It is not clear how the authors distinguish between the drug release because of diffusion from the micelles and degradation of the micelles in the drug delivery studies. Given the premise of the paper is a degradable platform, this is an important distinction to make –

Although the degradation product of HUBS are generally known to be non-cytotoxic, can the authors comment on the degradation products generated from the specific synthesized molecules used in this study? Can the authors comment on the cytotoxicity of the same?

Can the authors discuss the release rate of PTX from the micelles at 48 hours and the degradation of the micelles at 48 hours and address the discrepancy between the two?

Author Response

The study looks at the synthesis a of 4 hydrolyzable polyurea with different chemical structures to enable drug delivery via diffusion . The molecules were synthesized by a reaction between the diisocyanate and diamine monomers and extensively characterized .The ability of one of the polymers to deliver paclitaxel (PTX) in a murine model, both in vivo and in vitro is studied. HUB-based polymers are a promising route for degradable biomaterials and the authors have well- characterized the synthesized monomers: however, there are few issues that need to be addressed before the paper can be published

Response: Thanks very much for your positive comments on our work. We have conducted a complete revision regarding the issuesyou pointed out.

1. It is not clear how the authors distinguish between the drug release because of diffusion from the micelles and degradation of the micelles in the drug delivery studies. Given the premise of the paper is a degradable platform, this is an important distinction to make

Response: Thanks very much for your suggestion. We added the Korsmeyer Peppas equations in the manuscript as follows:

“Drug release was a complex process [15]. To simply explain the nature of drug release behaviors, a classic empirical equation was established by Peppas et al. [16]. The eqs. were written as:

(3)

(4)

In eq (3) and (4), Mt and M were the cumulative drug release at time t and infinite time, respectively; k was the proportionality constant, and n was the release exponent that it was related to the release mechanism of payloads. In the study of drug release, the increase in n indicated that the release was more influenced by a swelling controlled way. n was calculated using the eqs. (3) and (4), and the values of pH 6.8 and 7.4 were 0.32 and 0.21, respectively. The values of n were larger at pH 6.8 than pH 7.4, which was attributed to the faster hydrolysis of polyurea in an acidic environment.’’

2. Although the degradation product of HUBS are generally known to be non-cytotoxic, can the authors comment on the degradation products generated from the specific synthesized molecules used in this study? Can the authors comment on the cytotoxicity of the same?

Response: Thanks a lot for your helpful advice. We added relevant content in the manuscript as follows:

“PUM with HUBs was synthesized by the reversible reaction between cyclohexyl diisocyanate and tert-butyl diamine. With the hydrolysis of PUM, degradation products were cyclohexane-1,3-diyldimethanamine and tert-butyl diamine.”

After incubation with PUM at the concentration of 100.0 μg mL–1 for 72 h, the viability of L929 cells was kept around 93%, indicating negligible toxicity of PUM to normal cells.”

3. Can the authors discuss the release rate of PTX from the micelles at 48 hours and the degradation of the micelles at 48 hours and address the discrepancy between the two?

Response: Thank you for your reminding. The Mn of PEGylated PUM was reduced to 71.5%, but at 48 h, over 65% of PTX was released. The release rate of PTX from the PUM/PTX was significantly faster than the hydrolytic rate of PUM. The discrepancies between the two were because the release behaviors were influenced by the superposition of diffusion and hydrolysis.

Round 2

Reviewer 2 Report

Accept